# Selective Laser Melted M300 Maraging Steel—Material Behaviour during Ballistic Testing

**DOI:** 10.3390/ma14102681

**Published:** 2021-05-20

**Authors:** Ireneusz Szachogłuchowicz, Bartosz Fikus, Krzysztof Grzelak, Janusz Kluczyński, Janusz Torzewski, Jakub Łuszczek

**Affiliations:** 1Faculty of Mechanical Engineering, Institute of Robots & Machine Design, Military University of Technology, 2 Gen. S. Kaliskiego Street, 00-908 Warsaw, Poland; ireneusz.szachogluchowicz@wat.edu.pl (I.S.); krzysztof.grzelak@wat.edu.pl (K.G.); janusz.torzewski@wat.edu.pl (J.T.); jakub.luszczek@wat.edu.pl (J.Ł.); 2Faculty of Mechatronics, Armament and Aerospace, Institute of Armament Technology, Military University of Technology, 2 Gen. S. Kaliskiego Street, 00-908 Warsaw, Poland; bartosz.fikus@wat.edu.pl

**Keywords:** additive manufacturing, selective laser melting, ballistic testing, M300 steel, structural analysis

## Abstract

Significant growth in knowledge about metal additive manufacturing (AM) affects the increase of interest in military solutions, where there is always a need for unique technologies and materials. An important section of materials in the military are those dedicated to armour production. An AM material is characterised by different behaviour than those conventionally made, especially during more dynamic loading such as ballistics testing. In this paper, M300 maraging steel behavior was analysed under the condition of ballistic testing. The material was tested before and after solution annealing and ageing. This manuscript also contains some data based on structural analysis and tensile testing with digital image correlation. Based on the conducted research, M300 maraging steel was found to be a helpful material for some armour solutions after pre- or post-processing activities. Conducted solution annealing and ageing increased the ballistic properties by 87% in comparison to build samples. At the same time, the material’s brittleness increased, which affected a significant growth in fragmentation of the perforated plate. According to such phenomena, a detailed fracture analysis was made.

## 1. Introduction

Military usage of AM has become increasingly popular because of the significant growth of different technologies and dedicated materials for those technologies [1,2,3,4]. Until now, AM technologies mainly were used in military solutions for rapid prototyping (RP), and rapid tooling (RT) to a limited extent [5,6,7,8]. Recently, the popularity of using AM technologies in military solutions has increased, especially from the applicability point of view. One example is Kristoffersen et al.’s [9] research results, where ballistic perforation resistance of AM AlSi10Mg aluminium plates were taken into account. In comparison to AM materials and conventionally made materials, the authors determined that their ballistic appearance was negligible. The authors registered residual velocity for die-cast plates at a value of 230.7 m/s, while the AM was 231.6 m/s. The obtained results are very encouraging, especially considering that lightweight materials (aluminium and titanium-based) are an interest of many research facilities [10,11,12,13,14,15,16,17,18,19]. A different approach was suggested by Zochowski et al. [20], where the authors analysed ballistic properties of bulletproof vest inserts made of AM-made titanium alloy. Their research results indicated a significant capability of absorbing and dissipating the projectile impact energy from about 500 J to 0 J in 0.15 s. In Hassanin et al.’s [21] work, the authors proved an increased, higher ballistic performance of AM metamaterial characterised by the shape memory, superelasticity, and negative Poisson’s ratio properties compared to conventional steel armours. They improved the energy absorbed by unit mass from 91.28 and 252.35 N·mm/g in the case of solid steel and solid NiTi, respectively, to 495 N·mm/g for the optimized NiTi, AM auxetic structure.

Different usage of AM in military solutions was suggested by Limido et al., where the authors attempted to design a warhead penetrator using lattice structures to reduce its mass and guarantee, at the same time, its mechanical strength and performances, or even improve them. In their research, austenitic steels were considered. From a ballistic resistance point of view, ceramic materials were also prevalent. In the case of that type of material (mostly alumina–Al_2_O_3_), AM is possible. Jones et al. [22] performed mechanical and ballistic investigations of AM alumina-based armour plates. Unfortunately, their research indicated weaknesses of AM materials made using direct-ink writing technology (DIW), characterised by decreased ballistic properties by 45% as compared to isostatic pressed (IP) equivalent material. Similar research was conducted by Appleby-Thomas et al. [23], where the authors found that AM ceramic material was slightly less effective than that conventionally made. During the literature review, it was difficult to find research papers connected with the ballistic performance of steel parts obtained by AM. It is well known that those materials are commonly used in armoured parts production [24,25,26,27]. One example is maraging steels, which could be used in that type of solution. Kaiser et al. [28] found that that type of alloy met the acceptance standards of military specifications for First Article Certification and was certified for use on USA produced armoured systems. Regarding the usefulness of maraging steels in armour application and AM possibilities from a geometrical complexity point of view, ballistic tests were carried out. Additionally, it is well-known that maraging steels are amenable to heat treatment (solution annealing and ageing). Hence, two types of material were tested before and after heat treatment. Regarding our own previous research [29,30,31,32,33,34,35], to better understand AM material behaviour, it is essential to perform structural analysis of the material with properly described fractures. This approach is suggested in this paper, which will be helpful in understanding the effects of solution annealing and ageing in AM plates on tensile-deformation and ballistic-testing properties.

## 2. Materials and Methods

### 2.1. Material

The material used for samples manufacturing was gas atomized M300 maraging steel (Carpenter Additive, Philadelphia, PA, USA). Our own scanning electron microscopy (SEM) observations (Figure 1) indicated that powder particles were characterised by spherical shapes (diameters of 20–63 µm) with some satellites on the external surface.

Material chemical composition based on the producer’s quality control measures is shown in Table 1.

### 2.2. The Manufacturing Process

The SLM 125 HL (SLM Solutions, Lubeck, Germany), with a 400 W laser source, was used for samples manufacturing. Regarding the simple sample shape, its geometry was prepared directly in the software for process preparation-Magics (Materialise, Leuven, Belgium, version 19). An AM process was carried out under typical conditions in laser powder bed fusion (L-PBF) in an argon atmosphere, where the oxygen content was below 0.1%. We used the process parameters listed below to produce samples:Laser power: 300 WExposure velocity: 720 mm/sHatching distance: 0.12 mmLayer thickness: 0.05mm

The above mentioned parameters (hatching distance and layer thickness) are shown in Figure 2. The usage of the parameters, as mentioned earlier, values generated energy density at the level of 69.44 J/mm^3^, which is consistent with the equation:(1)$$\rho_{e}=\frac{L_{p}}{e_{v}h_{d}l_{t}}$$
where:L_P_–laser power (W);e_v_–exposure velocity (mm/s);h_d_–hatching distance (mm); l_t_–layer thickness (mm).

For ballistic tests, the samples had a cuboidal shape in the dimension of 110 mm × 110 mm × 6.5 mm (L × W × H). The height of the sample was selected to allow part penetration during the STANAG 4569 level I test (5.56 mm × 45 mm NATO projectile (SS109) at 30 meters with a velocity of 910 m/s) and a lack of penetration after a 20% projectile velocity decrease. This approach was suggested to better to understand material behaviour during dynamic loading in ballistic testing. The plate sample, shown in Figure 3, was made in such a way to allow for ballistic testing in a perpendicular direction according to the substrate plate surface (through the deposited material’s layers).

Visible cylindrical undercuts on the panel’s edges were necessary to allow substrate removal after the process was finished.

Half of the manufactured parts were heat-treated using a Nabertherm P300 annealing furnace (Nabertherm GmbH, Lilienthal, Germany). The process was divided into two stages: solution annealing performed under a temperature of 820 °C for one hour with air cooling, and ageing under a temperature of 480 °C with air cooling.

### 2.3. Description of the Testing Methodology

The ballistic tests were carried out using of the experimental stand presented in Figure 4. The stand consisted of a barrel launching system, calibre 5.56 mm (Figure 4g), and a specimen mount (Figure 4a) covered by a protective box with a bullet catcher (Figure 4d). To provide accurate values of impact velocity and to conduct the observation of terminal ballistics effects, a set of three devices were applied: an electromagnetic device for muzzle velocity measurement, EMG–1 (Figure 4f), an intelligent light screen, Kistler 2521A (Figure 4e) (Kistler, Switzerland), for bullet velocity measurement, and a high-speed camera, Phantom v1612 (Figure 4b) (Vision Research, Wayne, NJ, USA). Tests were conducted with 5.56 mm × 45 mm NATO M855 (SS109) steel core bullets. Following the results presented in [36], the expected value of the projectile velocity was equal to (935 ± 6) m/s regarding the specimen distance for the under–investigation ammunition and applied barrel length. Accounting for the required 20% decrease of impact velocity for the second conditions set, the round propellant mass was adjusted to obtain the velocity of approx. 750 m/s.

A Keyence VHX 7000 optical microscope (Keyence International, Mechelen, Belgium) was used for surface structural and fractures analysis. The material was mounted in resin, ground, polished, and etched to allow microstructural observations. Additional studies for determining powder grain shape and some microfracture phenomena were completed using a Jeol JSM-6610 SEM (JEOL Ltd., Tokyo, Japan) scanning electron microscope (SEM). To better describe material behaviour, tensile testing of samples was used. The tensile tests were carried out via the Instron 8802 (Instron, Norwood, MA, USA) testing machine. During those tests, digital image correlation (DIC) of the deformation process was made. DIC analyses, a non-contact optical technique to measure three-dimensional (3D) deformations, were made using Dantec Dynamics (Dantec, Ulm, Germany). The tensile testing rig with the DIC system is shown in Figure 5.

To check how the material’s properties were changed after heat treatment, tensile tests were conducted on samples for which geometry was designed based on the ASTM E466 96 standard (Figure 6). For each material condition, five samples were tested.

## 3. Results and Discussion

### 3.1. Tensile and Structural Analysis

Structural and tensile analysis was necessary to justify the character of material behaviour after a deterioration in different conditions: before heat treatment (BHT), after heat treatment (AHT), and after the impact of a projectile with different velocities. Additionally, this type of analysis was helpful for further fracture description.

The images of the material microstructure in each condition are shown in Figure 7. Z-direction, visible in Figure 7, is a direction of the material’s layers growth during AM process.

The material condition after the AM process is characterised by a typical, layered structure with visible molten pools [37,38]. The microstructure is very fine and regular in each molten pool, as is shown in Figure 8. 

Heat treatment resulted in a more homogeneous and fine-grained structure. It should be noted that the martensite needles were short and dark, which contributed either to an increase in hardness or a decreased material plasticity [39,40].

Tensile testing results for each sample group are shown in Figure 9. After heat treatment, tensile strength increased by almost 45% but, at the same time, elongation decreased by 67%, which significantly changed material properties, especially regarding its behaviour during dynamic loading such as ballistics testing.

Based on the obtained tensile tests, all characteristic values were measured, calculated, and presented in Table 2. Values shown in the Table, as mentioned earlier, are averages with standard deviation, which was calculated based on five measurements for each sample group.

More advanced tensile analysis of the samples was possible by using a DIC system. To adequately describe material behaviour during tensile testing, images at a certain level were taken (Figure 10) for BHT samples and for (Figure 11) AHT samples.

In the conventional yield point, the distribution of strain fields was very even. The sample showed no defect connected with the manufacturing process. In the case of material strain at the ultimate tensile strength point, the areas of maximum strain were marked with oval lines. There was a banding effect visible, which is evenly distributed. Before sample destruction (Figure 10d), a significant deformation at the level of 5% revealed areas of potential sources of cracking. Their location and number might relate to part porosity, which is visible in Figure 7. Considering the DIC image after fracture (Figure 10e), deformation fields are visible despite the lack of load. The white-dotted line (Figure 10e) indicates relaxation by the fracture, which appeared as a strain banding.

The DIC images captured during tensile tests of samples in the AHT condition are shown in Figure 11. To maintain consistency, the form of the obtained results is presented the same as in Figure 10. In the conventional yield point, there were visible small areas with increased strain value. This phenomenon could be related to the increased porosity caused by additional heat treatment [32]. The distribution of that area (marked using black arrows) is randomly placed on the whole sample’s surface.

In the case of the DIC image at the UTS point, there were no zones of constant strain. Only small areas of increased strain values were visible (Figure 11, dashed line). After exceeding the UTS point, no necking formation was visible, which indicates the brittleness and low plasticity of the material. In the last part of the material fracture, the cracking process was very dynamic, which caused the scale in Figure 10e to go out of the range. The course of the fracture line, which is complex, may be related to the irregular porosity and brittleness of the material.

### 3.2. Ballistic Tests Results

The ballistic tests were conducted under the conditions summarized in Table 3. As can be seen, for the initially applied value of impact velocity, the heat-treated samples exhibited more effective ballistic protection despite the more brittle character of specimen damage (described further below).

The material behaviour during the perforation of each target plate is shown in Figure 12.

Figure 12 contains ballistic test results captured by a high-speed camera, which allowed for the determination of the perforation mechanism during projectile penetration.

There were many failure mechanisms for the sample in the BHT condition, which was penetrated by projectile with a velocity equal to 936 m/s (Figure 12a). The first one was the plugging mechanism, highlighted in Table 3 using a yellow arrow to show the cut-out plug. During that kind of mechanism, the fragmentation of the material appeared in the form of several pieces of material with additional reverse fragmentation (marked by a red arrow in Figure 12a) and the generation of numerous shards on both sides.

Typical material fragmentation occurred for the AHT sample penetrated by a projectile with a velocity of 918 m/s (Figure 12b). A yellow arrow marked severed material parts. Particular attention should be paid to the dust formed during the penetration of this sample (it was marked with a yellow and red rectangle in Figure 12b) as this is a typical phenomenon in armour steels.

In the case of a plate in BHT condition, penetrated by a projectile with a velocity of 746 m/s, there was no perforation registered. In this case, a dust cloud had formed on the side of the projectile impact. Several small parts of the material were also spotted (Figure 12c Step II, red oval). This proves the plasticity of the material and good dispersion of the impact energy compared with plates in the AHT condition.

Regarding the last sample in AHT condition penetrated by a projectile with a velocity of 743 m/s, there was also no perforation. As in the previous case, a dust cloud was generated, but with more significant dimensions. Additionally, a greater number of tiny debris were noticed (Figure 12d, red ovals). Their increased number could be caused by a higher hardness of the material and its internal cracking.

### 3.3. Fracture Analysis

After ballistic testing, all types of fractures were considered. Structural analysis was necessary to determine the character of material deterioration in different conditions: BHT, AHT, and after the impact of a projectile with different velocities. Results in each condition are shown in Table 4.

As shown in Table 4, all samples were characterised by a significant porosity (above 1%) which is a very undesirable phenomenon. The occurrence of several such pores relates to the high temperature of the previously exposed layer, which caused key-hole porosity generation after delivering into the material volume a significant amount of energy [41,42]. It should be noted that process parameters used in the process had default values for M300 steel, but the melted material volume was too large to draw the proper amount of heat.

In non-perforated samples (Table 4, no. 3 and 4, marked by blue ovals), a part of the projectile was connected with the plate during the impact.

During DIC analysis, it was suggested that porosity could be a source of material cracking, visible in the plate after projectile impact with 743 m/s velocities in AHT condition (Table 4, No.4). Cracks in the material, which appeared after the projectile impact, went through the pores. That phenomenon is hazardous because cracking through the porosity, with an increased brittleness of the material, would cause significant material fragmentation directly after impact.

In the case of perforated plates penetrated by a projectile with a higher velocity (920 m/s), additional macroscopic analysis was completed as shown in Figure 13.

In both cases (BHT and AHT plates), many features were visible and characteristic of high-hardness material perforations. Comparing both cases, the material plate in the BHT condition seems to be more plastic than the AHT plates. The projectile’s lead cap caused a slight deformation on the entry side and left a small amount of its material on the external perforation surface. On the other side, a plug with a diameter like the perforation size was pushed out by the projectile. The surface of the perforation, shown in Figure 14 (areas a), was characterised by a significant deformation, which indicates the material’s ductility in that area. This phenomenon might cause a local increase in temperature during perforation due to friction between the surface of the projectile and the panel. That kind of approach was suggested by Kristoffersen et al. [9]. However, this issue requires more detailed research and will be a topic for further research. Additionally, there were cracks visible around the contact of the projectile and the plate, which indicates a material tendency for delamination (samples were manufactured layer-by-layer and perpendicular to the surface of the layer was the projectile impact) after such dynamic loading.

In the case of the ATH panel, the perforation channel on the projectile entrance side had an irregular shape due to the brittle tearing of the material which appeared as short and straight crack paths. Further penetration mechanisms caused a radial fracture and detachment of a large material part several times larger area than the projectile’s cross-sectional area. That phenomenon also caused material fragmentation. However, the fragmentation of the material does not indicate its ballistic resistance as it is only a result of its low ductility. In the cross-section of the perforation channel, a horizontal border was visible (Figure 14, areas b) from which the radial tearing of the material began. It indicates an interlayer cracking mechanism which may be caused by the layered structure of the AM material.

## 4. Conclusions

Obtained research results were helpful to understand AM M300 steel behaviour during ballistic testing conditions. A broad scope of the research with material conditions of BHT, AHT, and two projectile velocity levels, allowed us to draw the following conclusions:Despite the high UTS of M300 steel, it was possible to reduce projectile velocity only by 25% after perforation, with total material penetration;Solution and ageing annealing allowed for significantly increased ballistic properties. The same plate after heat treatment reduced a projectile velocity by 87%, unfortunately also with penetration. Additionally, heat treatment caused an increase in material fragmentation after impact;The source of material deterioration was twofold. On the one hand, it was connected with high material brittleness (especially after heat treatment), and on the other hand, it was connected with increased porosity of the tested plates;To reduce porosity in parts characterized by significant volume, some experimental process parameters selection, thermodynamical modelling of the exact geometry manufacturing, or hot isostatic pressing (HIP) annealing could be used;Despite the revealed weaknesses of the AM M300 maraging steel, this kind of material could be useful in some armour solutions, especially for the production of complex shapes of armoured parts. However, to properly manufacture each part, some pre- or post-process activities are necessary.

## Figures and Tables

**Figure 1 materials-14-02681-f001:**
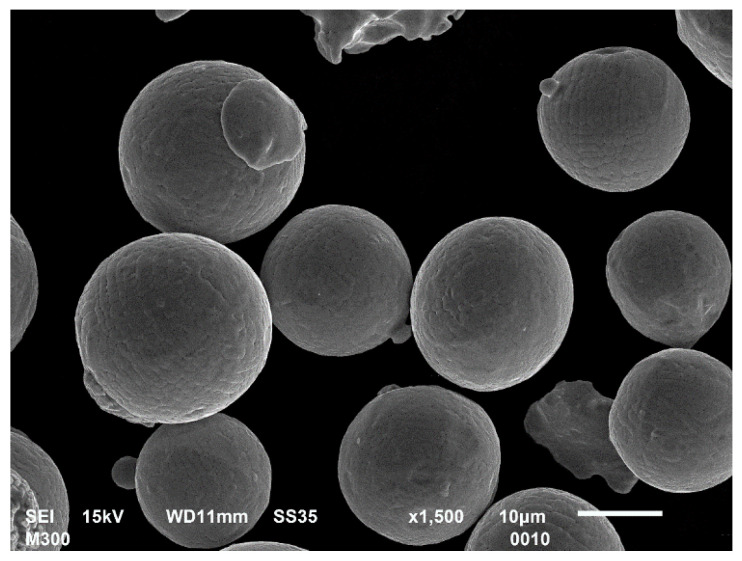
SEM image of M300 powder grains.

**Figure 2 materials-14-02681-f002:**
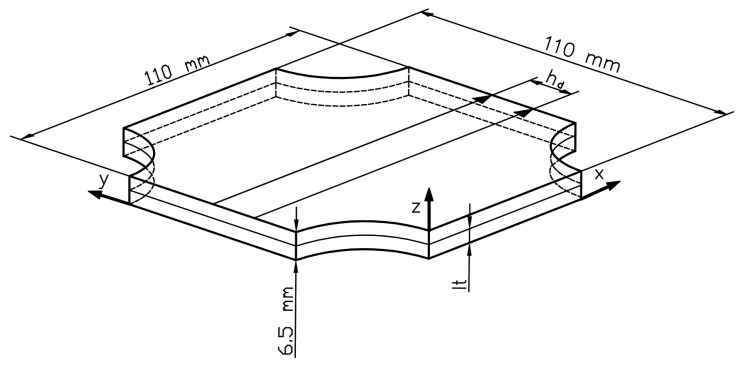
Explanation of layer thickness, hatching distance values, and sample plate dimension.

**Figure 3 materials-14-02681-f003:**
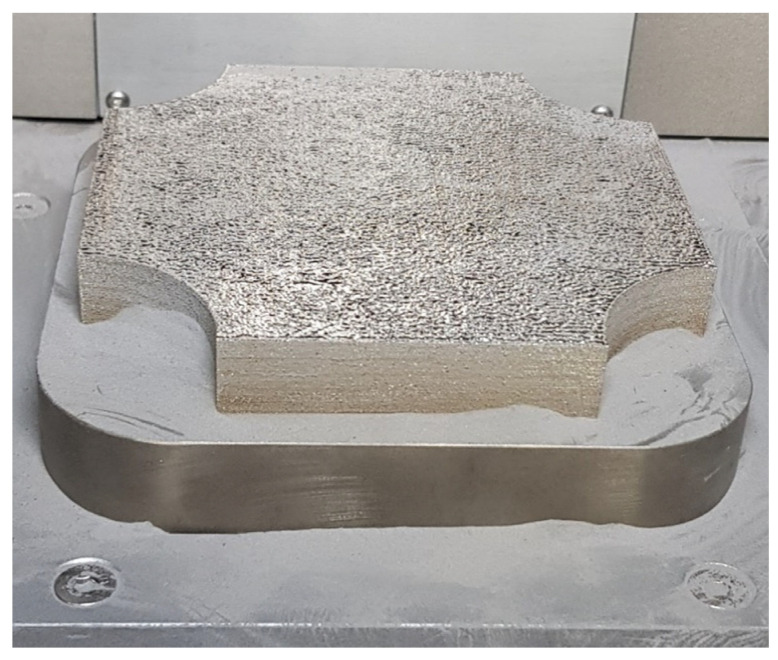
Sample panel on the SLM 125 HL’s substrate plate.

**Figure 4 materials-14-02681-f004:**
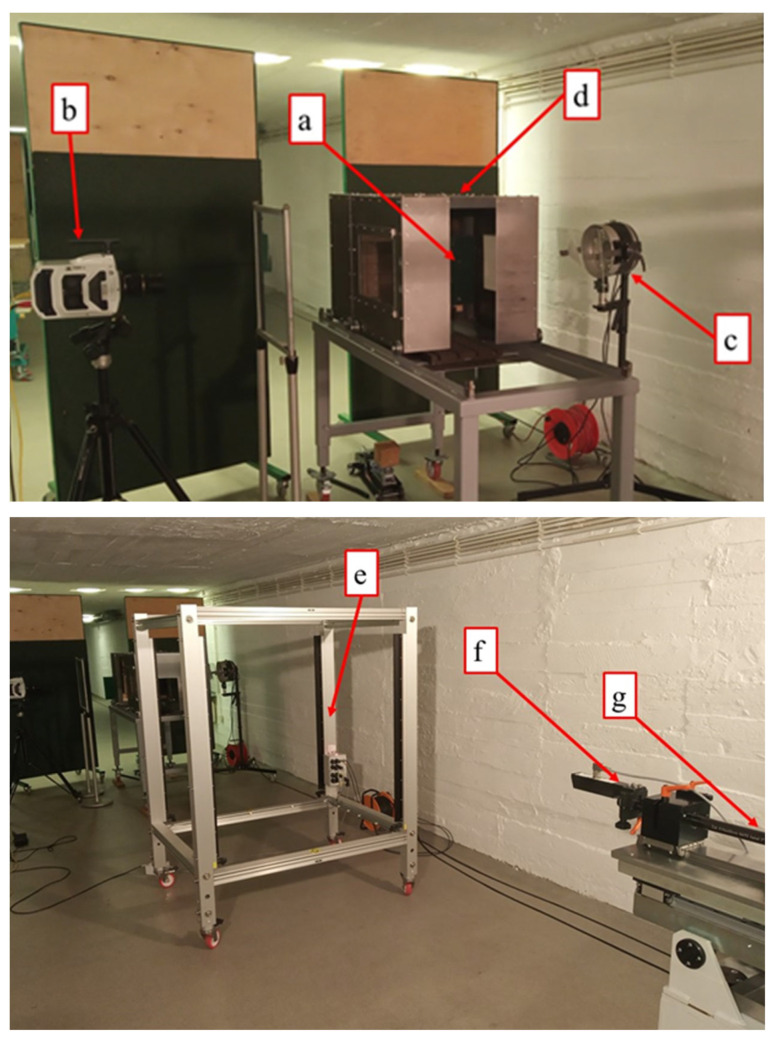
Experimental stand for terminal ballistics tests: (**a**) investigated specimen mount, (**b**) high-speed camera, (**c**) light source, (**d**) protective box with bullet catcher, (**e**) intelligent light screen, (**f**) electromagnetic muzzle device, and (**g**) barrel launching system, calibre 5.56 mm.

**Figure 5 materials-14-02681-f005:**
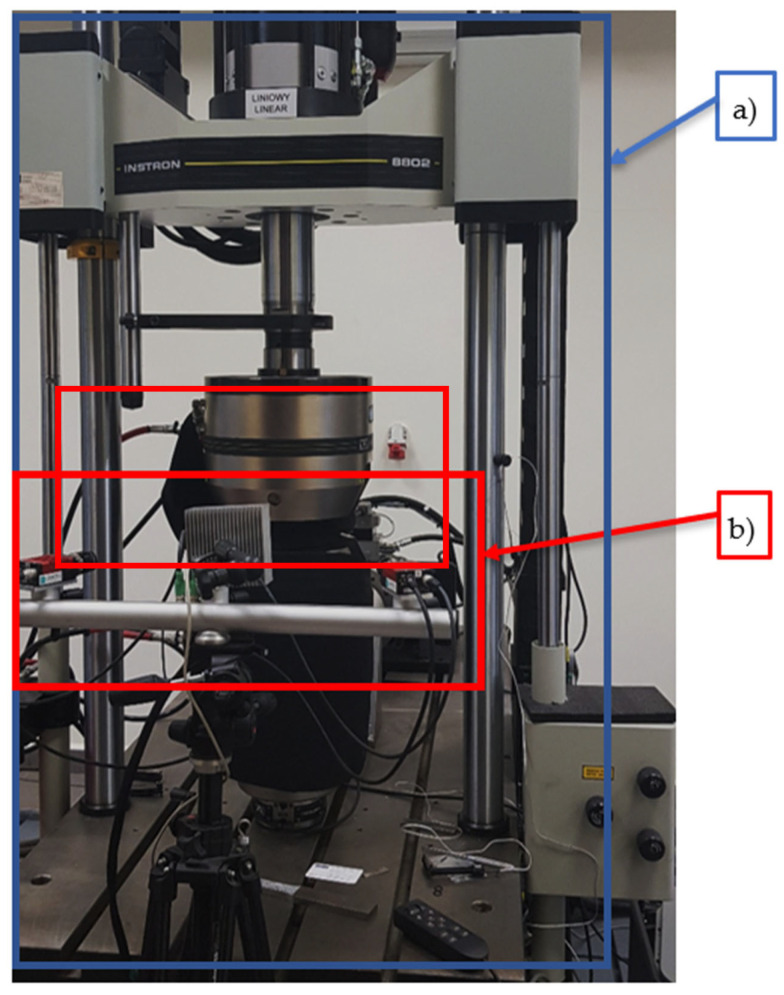
Servo-hydraulic pulsator for tensile tests Instron 8802 (**a**) with digital image correlation system DIC (**b**).

**Figure 6 materials-14-02681-f006:**
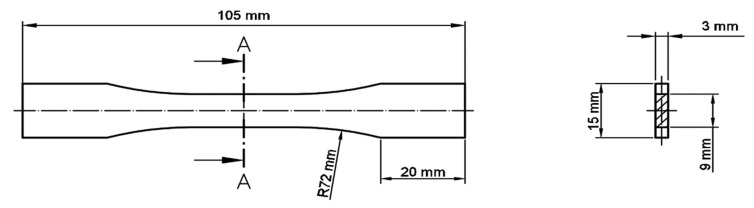
Sample dimensions basing on the ASTM E466 96 standard.

**Figure 7 materials-14-02681-f007:**
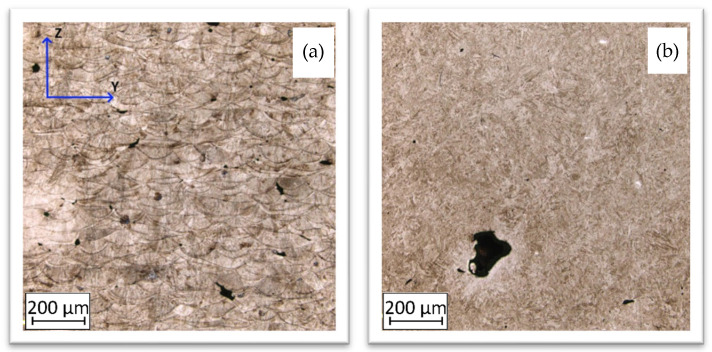
Microstructure of the AM M300 steel in BHT (**a**) and AHT (**b**) conditions.

**Figure 8 materials-14-02681-f008:**
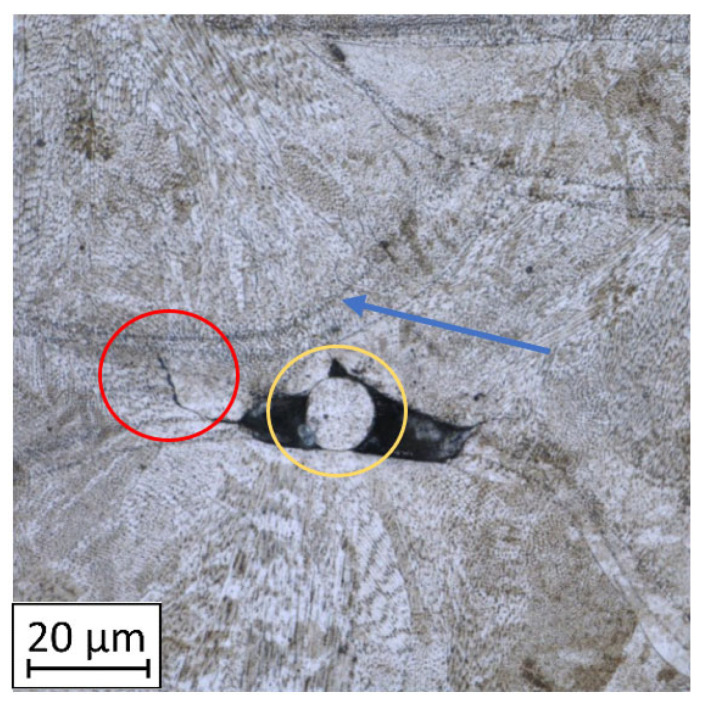
Microstructure of the AM M300 in BHT condition with marked crack (red circle) from non-melted grain (yellow circle) and visible melt-pool boundary (blue arrow).

**Figure 9 materials-14-02681-f009:**
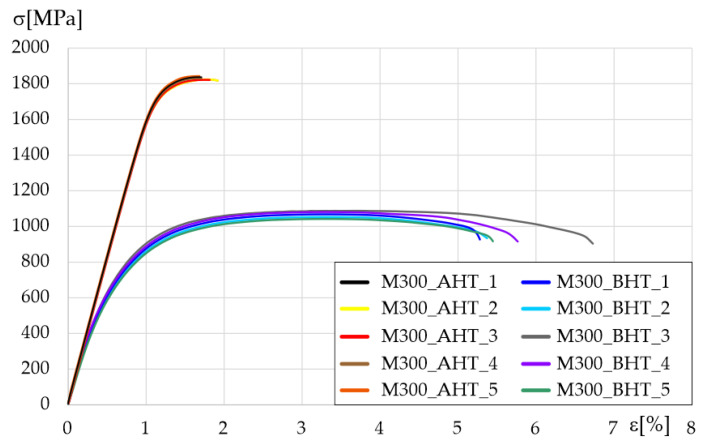
Tensile test results of M300 steel in AHT and BHT conditions.

**Figure 10 materials-14-02681-f010:**
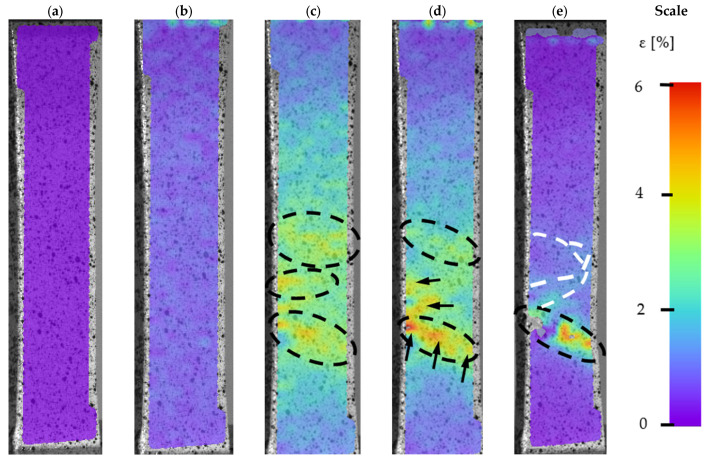
DIC of M300 steel in BHT condition during tensile testing, (**a**) a reference sample before the tensile testing, (**b**) Rp 0.2% proof strength, (**c**) the ultimate tensile strength (UTS) point, (**d**) breaking point, and (**e**) a sample after a fracture.

**Figure 11 materials-14-02681-f011:**
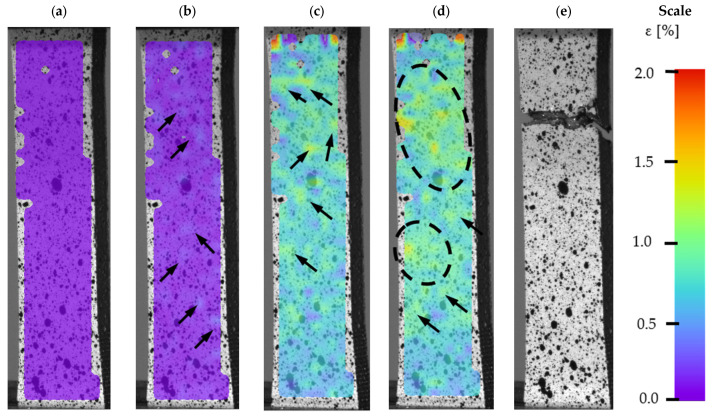
DIC of M300 steel in AHT condition during tensile testing, (**a**) a reference sample before the tensile testing, (**b**) Rp 0.2% proof strength, (**c**) the ultimate tensile strength (UTS) point, (**d**) breaking point, and (**e**) a sample after a fracture.

**Figure 12 materials-14-02681-f012:**
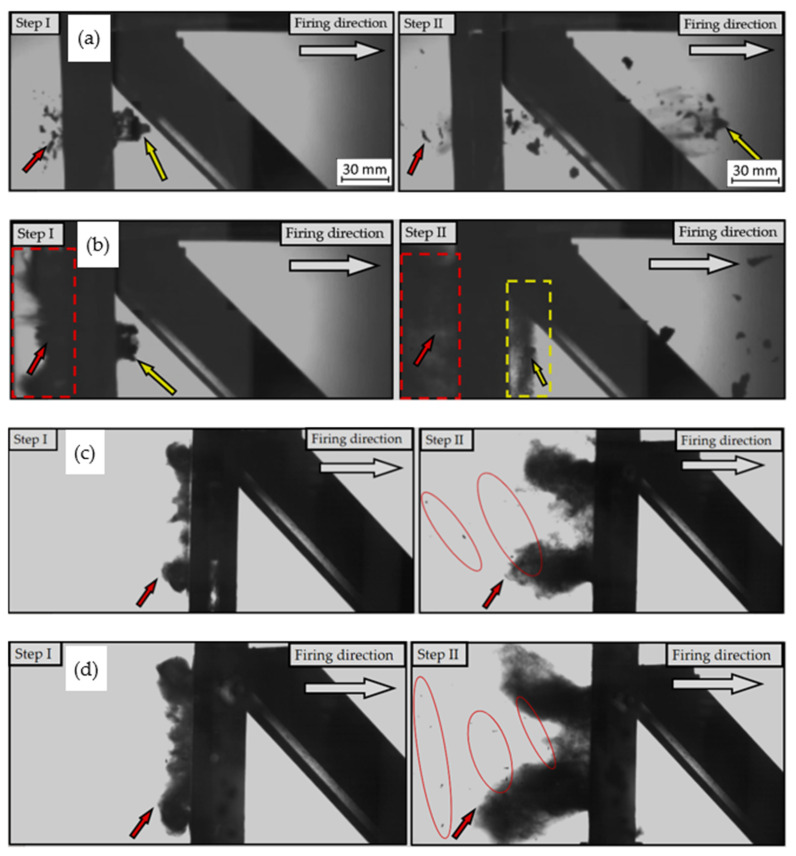
High-speed camera images captured during plates perforation: (**a**) BHT condition, 936m/s impact velocity; (**b**) AHT condition, 918m/s impact velocity; (**c**) BHT condition, 746m/s impact velocity; (**d**) AHT condition, 743m/s impact velocity.

**Figure 13 materials-14-02681-f013:**
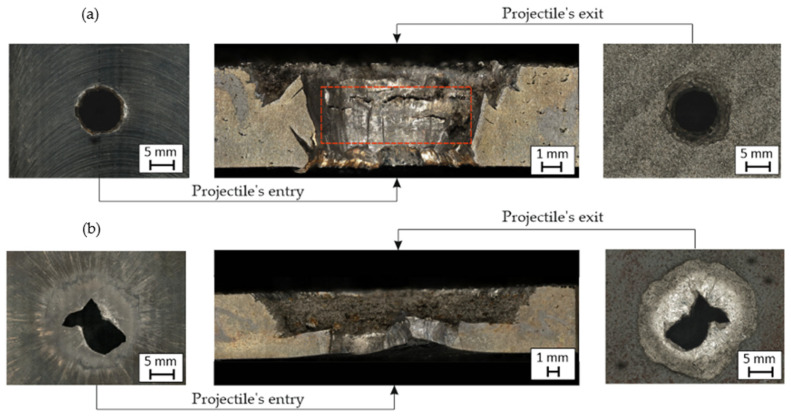
Fractures of projectile entry and exit of both penetrated plates: (**a**) BHT (**b**) AHT.

**Figure 14 materials-14-02681-f014:**
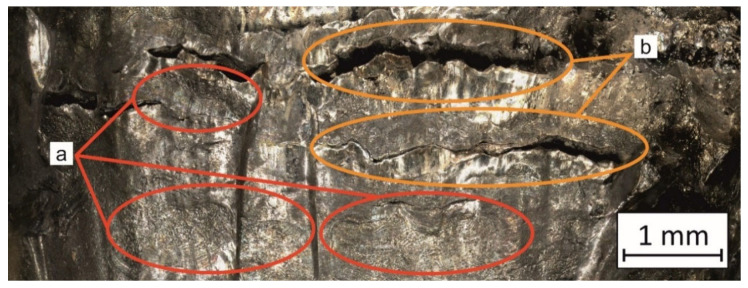
Perforation channel surface with visible deformed areas (**a**) and cracks (**b**) of the BHT plate.

**Table 1 materials-14-02681-t001:** M300 steel chemical composition.

Al	C	Cr	Co	Fe	Mn	Mo	Ni	N	O	P	Si	S	Ti	Cu
weight (%)														
0.07	0.02	0.06	8.97	balanced	0.03	4.82	18.43	0.01	0.02	0.005	0.05	0.002	0.98	0.1

**Table 2 materials-14-02681-t002:** Mechanical properties of BHT and AHT samples obtained during tensile testing.

Specimen Condition	Elasticity Modulus (GPa)	Rp_0.2%_ Proof Strength (MPa)	Ultimate Tensile Strength (MPa)	Elongation to Fracture (%)
BHT	156.3 ± 9.2	842.7 ± 3.1	1057.1 ± 5.3	5.84 ± 0.28
AHT	172.1 ± 2.9	1746.8 ± 1.6	1827.4 ± 2.1	1.99 ± 0.07

**Table 3 materials-14-02681-t003:** Conditions of experimental ballistic tests.

No.	Specimen Condition	Impact Velocity (m/s)	Impact Angle (°)	Residual Velocity (m/s)
**1**	BHT	936	90.8	696
**3**	AHT	918	91.1	120
**2**	BHT	746	90.6	No penetration
**4**	AHT	743	90.8	No penetration

**Table 4 materials-14-02681-t004:** Material structure before (BHT), after (AHT) heat treatment, and after deterioration with different velocities.

No.	Porosity (%)	Impact Velocity (m/s)	Material Condition
**1.**	3.86	936	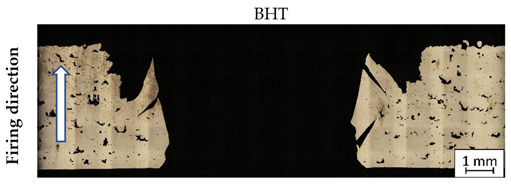
**2.**	1.94	918	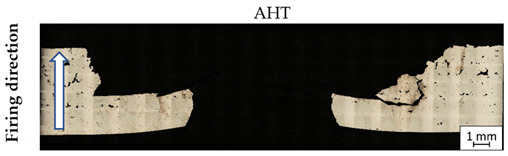
**3.**	3.69	746	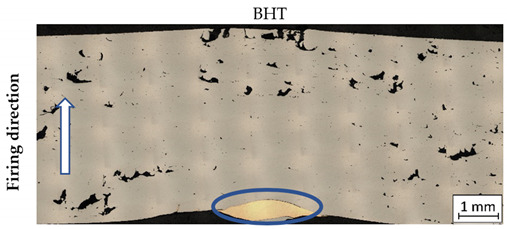
**4.**	2.21	743	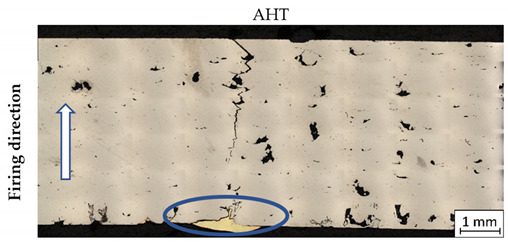

## Data Availability

Data sharing is not applicable.

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
