# Peer review of "Selective Laser Melted M300 Maraging Steel—Material Behaviour during Ballistic Testing"

_materials, 2021, doi:10.3390/ma14102681_

Round 1

Reviewer 1 Report

This paper investigated tensile-deformation and ballistic-testing properties of AM M300 maraging steel plates with and without solution annealing and aging. The solution annealing and aging changed AM sample properties significantly. The results are interested, but many mandatory revisions are required.  

Mandatories:

  1. Title: “Structural analysis” seems not to be suitable.
  2. Increased porosity in the AHT specimen seems to play a key rule in increase ballistic properties. Such structure with pores is mentioned to exhibit negative Poisson’s ratio and absorbing the impact energy in Introduction. Is the porosity in the AHT specimen is related to those features? Large pores in the AHT specimens and large number of small pores in the BHT specimens can be seen, while no evidence of the increased porosity in the AHT specimen was shown in the paper. In conclusion no.4, authors seem to want to reduce the increased porosity. If the porosity is the key point in the paper, the conclusion confused readers.

Following mandatories in ascending order of pagination:

  1. Abbreviation of “additive manufacturing” is defined, but the abbreviation is not often used in the text.
  2. Last sentence in Introduction, definition of “material’s deterioration process” in the paper is not clear. The paper shows effects of solution annealing and aging in the AM plates on tensile-deformation and ballistic-testing properties.
  3. In the last paragraph in page 3, “STA-NAG 4569 – level I” should be explained in the text. In addition, “to allow fatigue testing in a perpendicular, …” is mentioned, but no results are shown in the paper.
  4. In the last paragraph in page 4, “5.56 mm” must be diameter in “caribre 5.56 mm” and “5.56 mm M855 …”, but it was not mentioned. How is adding “Φ” before “5.56 mm” (figure caption in Fig. 3 as well)? In addition, the unit of “length” is not “m/s” in “… barrel length is equal to (935 +- 6) m/s”, “length” must be “velocity”.
  5. “non-melted grain” in the figure 6 caption is difficult to recognize in figure 6. Please show which the non-melted grain (maybe not grain but a part of a powder) in Fig.6.
  6. A paragraph “Heat treatment … decreased material plasticity” in page 7 explain which figure?
  7. “It can be noticed … the martensite … precipitated carbides …”, In general, precipitation of the maraging steel is not carbides (probably intermetallic compounds depending on constituent elements). What does “which” show in the sentence?
  8. Indication of “BHT” and “AHT” in figure 7, “AHT” in figure caption (Fig. 8) and “BHT” in figure caption (Fig. 9) are not correct indication.
  9. Specified levels described in the last paragraph of page 8 should be add in the figure captions in figures 8 and 9.
  10. In page 9, “deformation” should be “strain (ε)”, since figures 8 and 9 show “strain”. “figure 3” and “Fig. 1e” are not correct indication. Probably, figure 8c and Fig. 8e, respectively. The white dotted line in figure 8e indicate relaxation by the fracture in the large strain bandings. “The kind of phenomenon relates to the plastic deformation” is not clear.
  11. “ increased material porosity” is not clear, as mentioned in 2.
  12. “Without heat treatment” and “Heat treated” in table 2, while “Before heat treatment” and “After heat treatment” in tables 3 & 4, like “BHT” and “AHT” in the text. In addition, display order of samples and “impact velocity” in table 2 is different from those (“projectile velocity”) in tables 3 & 4.
  13. Although “table 3a)”, “table 3b)", “table 3c)”, “table 3d)” are mentioned in the text, but neither “a)”, “b)”, “c)”,nor “d)” in table 3.
  14. In table 4, which plane (upper of bottom) was hit by a projectile? It should be mentioned. In addition, a portion with different contrast is seen to be formed in a lens shape at a bottom plane at projectile velocities of 746 and 743 m/s, which is not mentioned in the text. What are they?
  15. In the first sentence in page 13, it seems to be mentioned regarding AHT specimens, but not mentioned which ones in table 4. Also said “undesirable”, however, the AHT specimens exhibited superior ballistic resistance. Is the increased porosity not associated with ballistic resistance? Following sentences show AM process conditions, but the increased porosity authors mentioned is produced by solution annealing and aging. I am afraid the paragraph is not appropriate explanation.
  16. Is the detachment of a large materials part in the AMT specimen is not associated with increased porosity? Aging makes the materials hard and less ductility. The large detachment (fracture) can absorb large impact energy. This can be associated with high yield stress and porosity. 
  17. The paper focus on ballistic resistance, and the resistance is larger in the AHT specimen. I am not sure the reason why the perforation surface of the AHT specimen is not shown, although that of BHT specimen is shown.

Author Response

Dear Reviewer,

In the beginning, we would like to thank you for your valuable comments. We made proper corrections based on your comments. All changes made in our manuscript which were made according to your comments were yellow-highlighted. Regarding each paragraph:

  1. Title: “Structural analysis” seems not to be suitable.

Answer: We changed a title, and now it is as follows: “Selective laser melted M300 maraging steel – material behaviour 0after ballistic testing”

  1. Increased porosity in the AHT specimen seems to play a key rule in increase ballistic properties. Such structure with pores is mentioned to exhibit negative Poisson’s ratio and absorbing the impact energy in Introduction. Is the porosity in the AHT specimen is related to those features? Large pores in the AHT specimens and large number of small pores in the BHT specimens can be seen, while no evidence of the increased porosity in the AHT specimen was shown in the paper. In conclusion no.4, authors seem to want to reduce the increased porosity. If the porosity is the key point in the paper, the conclusion confused readers.

Answer: We cannot agree with your comment. Regarding ballistic resistance, porosity has always a negative influence on that kind of properties – it is a potential cracking spot (that is why all armor plates are densified as much as possible). The issue which you mentioned relates to energy dissipation which is a quite different factor than ballistic resistance. As was mentioned at the end of the introduction – our main goal was to describe the material’s behaviour during ballistic impact.  That is why we mentioned HIP in the conclusions – it could increase the ballistic resistance of the part.

Following mandatories in ascending order of pagination:

  1. Abbreviation of “additive manufacturing” is defined, but the abbreviation is not often used in the text.

Answer: We made proper corrections – the word additive manufacturing was changed on an abbreviated one.

  1. Last sentence in Introduction, definition of “material’s deterioration process” in the paper is not clear. The paper shows effects of solution annealing and aging in the AM plates on tensile-deformation and ballistic-testing properties.

Answer: You are 100% right – we wanted to use a shorter word, but if it is not clear in your opinion, we changed it to a version which you suggested.

  1. In the last paragraph in page 3, “STA-NAG 4569 – level I” should be explained in the text. In addition, “to allow fatigue testing in a perpendicular, …” is mentioned, but no results are shown in the paper.

Answer: STANAG 4569 is a standard, as well as ASTM E466 96 standard for tensile testing. That is why we decided to put only that kind of information. Anyway, we put specified conditions of STANAG 4569 – level 1 in the bracket directly after the standard name. Regarding the fatigue issue, it was a mistake – now it is rewritten.

  1. In the last paragraph in page 4, “5.56 mm” must be diameter in “calibre 5.56 mm” and “5.56 mm M855 …”, but it was not mentioned. How is adding “Φ” before “5.56 mm” (figure caption in Fig. 3 as well)? In addition, the unit of “length” is not “m/s” in “… barrel length is equal to (935 +- 6) m/s”, “length” must be “velocity”.

Answer: Regarding descriptions of the ammunition, there is no need to put “Φ” before the value of caliber of barrel applied for launching of considered projectiles. What is important, the "caliber" is the term describing the barrel - not bullet (bullet diameter differs from caliber). Considering this fact, we added "caliber" in paragraph concerning the experimental stand (barrel) description. In the corrected manuscript we applied the full name (according with the nomenclature) of investigated cartridges (5.56x45 mm NATO M855 (SS109) ) to describe used ammunition clearly. In the second part, please have a look at the whole sentence, because in the beginning is: “Following the results presented in [30], the expected value of the projectile velocity (…)”. Indeed, the form of this sentence is complex and could be not clear, so we have rephrased it to avoid such misunderstandings for other readers.

  1. “non-melted grain” in the figure 6 caption is difficult to recognize in figure 6. Please show which the non-melted grain (maybe not grain but a part of a powder) in Fig.6.

Answer: We put an additional yellow circle and point to the unmelted grain which caused a pore generation.

  1. A paragraph “Heat treatment … decreased material plasticity” in page 7 explain which figure?

Answer: This sentence is not connected with a figure but with a natural phenomenon caused by martensite generation – we put additional citations directly after the sentence.

  1. “It can be noticed … the martensite … precipitated carbides …”, In general, precipitation of the maraging steel is not carbides (probably intermetallic compounds depending on constituent elements). What does “which” show in the sentence?

Answer: We decided to remove the word “carbide” because it could be not clear enough. The sentence has been rewritten.

  1. Indication of “BHT” and “AHT” in figure 7, “AHT” in figure caption (Fig. 8), and “BHT” in figure caption (Fig. 9) are not correct indication.

Answer: You are absolutely right, thank you. We corrected it.

  1. Specified levels described in the last paragraph of page 8 should be add in the figure captions in figures 8 and 9.

Answer: We put deformations level as you suggested has been put into the figure 10 and 11 captions.

  1. In page 9, “deformation” should be “strain (ε)”, since figures 8 and 9 show “strain”. “figure 3” and “Fig. 1e” are not correct indication. Probably, figure 8c and Fig. 8e, respectively. The white dotted line in figure 8e indicate relaxation by the fracture in the large strain bandings. “The kind of phenomenon relates to the plastic deformation” is not clear.

Answer: We replaced “deformation” with “strain” in the case of DIC descriptions. Thank you for your comment about figures – we made it right. Regarding your comment about a white dotted line in figure 10e we decided to put your suggestion and remove our sentence which in your opinion could be not clear.

  1. “ increased material porosity” is not clear, as mentioned in 2.

Answer: The word “material” has been removed. Additionally, we put additional information about the reason for porosity increasing and cover it by own previous research results.

  1. “Without heat treatment” and “Heat treated” in table 2, while “Before heat treatment” and “After heat treatment” in tables 3 & 4, like “BHT” and “AHT” in the text. In addition, display order of samples and “impact velocity” in table 2 is different from those (“projectile velocity”) in tables 3 & 4.

Answer: We put BHT and AHT in all tables as you suggested. Also, we unified all descriptions and orders in tables.

  1. Although “table 3a)”, “table 3b)", “table 3c)”, “table 3d)” are mentioned in the text, but neither “a)”, “b)”, “c)”,nor “d)” in table 3.

Answer: We put the same descriptions as in figure 12 – 12.1 – 12.4 and replace them in the text.

  1. In table 4, which plane (upper of bottom) was hit by a projectile? It should be mentioned. In addition, a portion with different contrast is seen to be formed in a lens shape at a bottom plane at projectile velocities of 746 and 743 m/s, which is not mentioned in the text. What are they?

Answer: We put additional arrows with information about firing direction. A part which has a different contrast is a piece of the projectile – we put a proper comment in the text and yellow-highlighted it.

  1. In the first sentence in page 13, it seems to be mentioned regarding AHT specimens, but not mentioned which ones in table 4. Also said “undesirable”, however, the AHT specimens exhibited superior ballistic resistance. Is the increased porosity not associated with ballistic resistance? Following sentences show AM process conditions, but the increased porosity authors mentioned is produced by solution annealing and aging. I am afraid the paragraph is not appropriate explanation.

Answer: We put additional information to easier recognize the exact specimen. As we mentioned before – porosities always decrease the ballistic performance of the material and it is an undesirable factor. We described how porosity affects the fracture mechanism – not a ballistic performance of the part. Indeed, heat treatment increased the ballistic performance of the material, but the fragmentation and cracking caused by porosities and increased brittleness are undesirable. We tried to describe fracture mechanism – not ballistic properties in that part.

  1. Is the detachment of a large materials part in the AMT specimen is not associated with increased porosity? Aging makes the materials hard and less ductility. The large detachment (fracture) can absorb large impact energy. This can be associated with high yield stress and porosity. 

Answer: Please have a look at Figure 12 (No. 1 and 2). A big part in a form of the plug has been detached from the plate (No.1) which is a BHT condition. At the same time in AHT (No.2) sample, there were a lot of parts with smaller dimensions. As it is mentioned in the text the plugging mechanism could be affected by a layered structure of AM material, but it must be a topic of some deeper analysis.  Indeed, aging makes the materials hard and less ductility, but we cannot point that it caused detachment of bigger parts of the AHT material, but the only bigger volume (a large group of significantly smaller parts) – which is visible in figure 10.

  1. The paper focus on ballistic resistance, and the resistance is larger in the AHT specimen. I am not sure the reason why the perforation surface of the AHT specimen is not shown, although that of BHT specimen is shown.

Answer: As we mentioned in conclusion 2. “Solution and ageing annealing allow increasing the ballistic properties significantly”, then we are one in such regard. Please look at figure 10 where both perforations are visible. We put an additional image of the BHT plate in figure 11 to discuss all phenomena which appear after the projectile’s impact on as-built AM plate. 

We hope our corrections and explanations meets your expectations. We did our best to fit our manuscript for your and other reviewers’ comments. Thank you very much for your valuable and helpful comments.

With kind regards,

Authors

Reviewer 2 Report

This is an experimental work with publication value, as there are very few studies published in the field. However, the paper requires improvements and amendments, to address the issues identified. Please refer to my (section by section) comments below.

Abstract:

Please rewrite to offer details on the findings (quantifying some of the most important)

  1. Introduction:

In many instances the statements made (from the reviewed literature) have to be supported by specific values or ranges of values, such as:

  • Unclear what negligible appearance is: ‘In comparison to AM-made materials and conventionally made materials, authors determined that their ballistic appearance was negligible.’
  • Please quantify significant: ‘Their research results indicated a significant capability of absorbing and dissipating the projectile impact energy.’
  • Please quantify higher performance: ‘In Hassanin et al.’s work authors proved increased higher ballistic performance of additively manufactured metamaterial’
  • Please quantify decreased properties: ‘Unfortunately, their research indicated weaknesses of AM materials characterized by decreased ballistic properties compared to conventionally made equivalent material.’

This statement contradicts the previously discussed literature review: ‘During the literature review, it was not easy to find research papers connected with the ballistic performance of steel parts obtained by additive manufacturing.’

Please add several papers here (since the claim made is very generic): ‘It is well known that those materials are commonly used in armoured parts production.’

A Table summarising the findings (with numerical values) is proposed to be added

  1. Materials and Methods:

Syntax has to be corrected/improved in several instances, such as:

  • ‘For samples manufacturing gas atomized M300 maraging (Carpenter Additive, Philadelphia, PA, USA) steel was used.’ This sentence should not start with ‘for’
  • ‘A process transaction had taken place under typical conditions in laser powder bed fusion (L-PBF)’ The sentence is unclear; what is ‘process transaction’? It is not a term used in AM (‘fabrication’ may be used instead)

A graph showing the manufacturing parameters listed section 2.2 (hatching space, layer thickness) should be added

A drawing showing the cuboidal specimen (geometry and dimensions) should be added

Figure 2 shows an image only, without any dimensions: ‘Samples with dimensions shown in Figure 2 were made in such a way to allow fatigue testing in a perpendicular, parallel, and angled direction (at an angle of 45 °) according to the plane parallel to the substrate plate surface.’ Also, is this (fatigue) specimen the same as the cuboidal specimen referred above?

A drawing showing the fatigue specimen (geometry and dimensions) should be added

Overall, section 2.2 needs to be more clear and it is suggested to be rewritten

Section 2.3 – page 5: Please add the letters designated / shown in Figure 3 in each of the sentences describing the testing apparatus and configuration

Figure 4: It is not clear what the a) and b) indicated items are (one cannot distinguish from this image what exactly is shown – i.e. a) shows the upper head of the tensile machine and b) a cable or something like that)

  1. Results and discussion

Section 3.1: The description of the material preparation for optical microscopy should be in section 2 (Materials and Methods). Please move this there.

Figure 6 is showing a pore or a defect. This is not described in the accompanying text (paragraph above the figure). Please add a description there, providing also the size of this defect/pore.

References (from the literature) should be added here or hardness measurements performed by the authors (if this statement refers to results obtain from their own analysis): ‘All these phenomena contribute either to an increase in hardness or decreased material plasticity.’

Again, these details (on experimental setup and standards used) should be in section 2 (Materials and Methods), please transfer these there: ‘To check how material’s properties were changed after heat treatment, tensile tests had been done on samples which geometry has been designed based on the ASTM E466 96 standard. For each material condition, five samples were tested.’

A table with the basic mechanical properties obtained from each of the five tests should be added (presenting only the stress-strain curves in an overlap form in Figure 7 is not sufficient or appropriate), namely the table should have the average of: Elasticity Modulus, yield strength, ultimate strength, elongation to fracture

Provide the definition of ‘conventional yield point’

What is ‘a moment just before the fracture’. Please quantify moment

The caption of Figure 8 is neither descriptive nor provides any information on what a) to e) subfigure is. Please add details (rewrite caption). The same applies to Figure 9 caption.

Table 3 should be a Figure (since it shows images). Also, a scale indicator should be added in the images. Also, ‘Table 3’ caption needs to describe in detail what one sees in each of the subfigures (these should be numbered also), similarly to what needs to be done in Figure 8 and 9 captions.

Have the authors performed a porosity measurement? It is apparent from various images (and also mentioned in the text) that the material exhibits a high level of porosity, though no values are provided. This needs to be added if available or a comment has to be made to clarify why this was not included in the analysis and how porosity can influence the results.

Again, Table 3 should be a figure – see also other comments on additions that have to be made to Table 3 (subfigures, caption etc), to apply these also here.

Author Response

# Reviewer 2

Dear Reviewer,

In the beginning, we would like to thank you for your valuable comments. We made proper corrections based on your comments. All changes made in our manuscript which were made according to your comments were green-highlighted. Regarding each section of your comments:

Abstract:

Please rewrite to offer details on the findings (quantifying some of the most important)

Answer: We have rewritten the abstract by removing the first part and adding some quantified final results.

  • Unclear what negligible appearance is: ‘In comparison to AM-made materials and conventionally made materials, authors determined that their ballistic appearance was negligible.

Answer: We put an additional sentence in the text: Authors registered residual velocity for die-cast plates in a value of 230.7 m/s and for the AM one 231.6 m/s.

  • Please quantify significant: ‘Their research results indicated a significant capability of absorbing and dissipating the projectile impact energy.’

Answer: The sentence with a short description has been extended by: “(…) the projectile impact energy from about 500J to 0J in 0.15s.

  • Please quantify higher performance: ‘In Hassanin et al.’s work authors proved increased higher ballistic performance of additively manufactured metamaterial’

Answer: We add a sentence: “They improved the energy absorbed by unit mass from 91.28 and 252.35 N·mm/g in the case of solid steel and solid NiTi, respectively, to 495 N·mm/g for the optimized NiTi, AM auxetic structure.”

  • Please quantify decreased properties: ‘Unfortunately, their research indicated weaknesses of AM materials characterized by decreased ballistic properties compared to conventionally made equivalent material.’

Answer: We extended this sentence by adding proper values from mentioned literature: “Unfortunately, their research indicated weaknesses of AM materials made using direct-ink writing technology (DIW) characterized by decreased ballistic properties in a value of 45% compared to isostatic pressed (IP) equivalent material.”

This statement contradicts the previously discussed literature review: ‘During the literature review, it was not easy to find research papers connected with the ballistic performance of steel parts obtained by additive manufacturing.’

Answer: Please note that in all mentioned literature, authors considered aluminum alloys and titanium alloys, that is why such a statement appeared.

Please add several papers here (since the claim made is very generic): ‘It is well known that those materials are commonly used in armoured parts production.’

Answer: We put four different papers where the ballistic performance of steels was considered.

24.Zhou, N.; Wang, J.; Peng, C.; Tong, Z. Experimental and numerical study on the ballistic resistance of steel-fibre reinforced two-layer explosively welded plates. Mater. Des. 2014, 54, 104–111.

  1. Atapek, S.H. Development of a new armor steel and its ballistic performance. Def. Sci. J. 2013, 63, 271–277.
  2. Wen, Y.; Xu, C.; Wang, H.; Chen, A.; Batra, R.C. Impact of steel spheres on ballistic gelatin at moderate velocities. Int. J. Impact Eng. 2013, 62, 142–151.
  3. Coles, L.A.; Roy, A.; Sazhenkov, N.; Voronov, L.; Nikhamkin, M.; Silberschmidt, V. V. Ice vs. steel: Ballistic impact of woven carbon/epoxy composites. Part I – Deformation and damage behaviour. Eng. Fract. Mech. 2020, 225, 106270.

A Table summarising the findings (with numerical values) is proposed to be added

Answer: It could be a very good idea if there will be the same values considered in each paper. Unfortunately, in the cited literature, the authors analyzed different factors. Sometimes ballistic tests were made only to create a numerical model. Additionally, our article is not a typical paper about ballistic analysis because we focused on material behaviour a lot. Comparing research results from literature in a table will not be meaningful. That is why we decided not to put such a table.

  1. Materials and Methods:

Syntax has to be corrected/improved in several instances, such as:

  • ‘For samples manufacturing gas atomized M300 maraging (Carpenter Additive, Philadelphia, PA, USA) steel was used.’ This sentence should not start with ‘for’

Answer: The sentence has been rewritten.

  • ‘A process transaction had taken place under typical conditions in laser powder bed fusion (L-PBF)’ The sentence is unclear; what is ‘process transaction’? It is not a term used in AM (‘fabrication’ may be used instead)

Answer: We mean that the process was run under exact conditions. Anyway, we rephrased it.

A graph showing the manufacturing parameters listed section 2.2 (hatching space, layer thickness) should be added

A drawing showing the cuboidal specimen (geometry and dimensions) should be added

Answer: We put a figure with process parameters explanation and sample plate dimensions.  

Figure 2 shows an image only, without any dimensions: ‘Samples with dimensions shown in Figure 2 were made in such a way to allow fatigue testing in a perpendicular, parallel, and angled direction (at an angle of 45 °) according to the plane parallel to the substrate plate surface.’ Also, is this (fatigue) specimen the same as the cuboidal specimen referred above?

Answer: Regarding the fatigue issue, it was put by mistake – now it is rewritten.

A drawing showing the fatigue specimen (geometry and dimensions) should be added

Answer: We put a geometry of samples used for tensile testing and DIC analysis in figure 6.

Overall, section 2.2 needs to be more clear and it is suggested to be rewritten

Answer: We shorten this section by removing not important data and rewriting some parts of the text.

Section 2.3 – page 5: Please add the letters designated / shown in Figure 3 in each of the sentences describing the testing apparatus and configuration

Answer: We made suggested corrections.

Figure 4: It is not clear what the a) and b) indicated items are (one cannot distinguish from this image what exactly is shown – i.e. a) shows the upper head of the tensile machine and b) a cable or something like that)

Answer: We create boxes to make it more visible.

  1. Results and discussion

Section 3.1: The description of the material preparation for optical microscopy should be in section 2 (Materials and Methods). Please move this there.

Answer: We moved this sentence to the suggested place.

Figure 6 is showing a pore or a defect. This is not described in the accompanying text (paragraph above the figure). Please add a description there, providing also the size of this defect/pore.

Answer: We put an additional envelope, and name it in the caption. To allow size analysis we put a scale in the bottom-left corner.

References (from the literature) should be added here or hardness measurements performed by the authors (if this statement refers to results obtain from their own analysis): ‘All these phenomena contribute either to an increase in hardness or decreased material plasticity.’

Answer: This is a natural phenomenon caused by martensite generation – we put additional citations directly after the sentence.

Again, these details (on experimental setup and standards used) should be in section 2 (Materials and Methods), please transfer these there: ‘To check how material’s properties were changed after heat treatment, tensile tests had been done on samples which geometry has been designed based on the ASTM E466 96 standard. For each material condition, five samples were tested.’

Answer: We moved to the suggested place.

A table with the basic mechanical properties obtained from each of the five tests should be added (presenting only the stress-strain curves in an overlap form in Figure 7 is not sufficient or appropriate), namely the table should have the average of: Elasticity Modulus, yield strength, ultimate strength, elongation to fracture

Answer: We put an additional table in the paper with mentioned values, we cannot put yield strength because of lack of visible Yield point, that is why we decided to put Rp 0.2% proof strength.

Provide the definition of ‘conventional yield point’

Answer: There was a mistake, we replace it with “Rp 0.2% proof strength”

What is ‘a moment just before the fracture’. Please quantify moment,

Answer: We replace this word with “breaking point”

The caption of Figure 8 is neither descriptive nor provides any information on what a) to e) subfigure is. Please add details (rewrite caption). The same applies to Figure 9 caption.

Answer: We rephrased a caption based on another reviewer’s comment.

Table 3 should be a Figure (since it shows images). Also, a scale indicator should be added in the images. Also, ‘Table 3’ caption needs to describe in detail what one sees in each of the subfigures (these should be numbered also), similarly to what needs to be done in Figure 8 and 9 captions.

Have the authors performed a porosity measurement? It is apparent from various images (and also mentioned in the text) that the material exhibits a high level of porosity, though no values are provided. This needs to be added if available or a comment has to be made to clarify why this was not included in the analysis and how porosity can influence the results.

Again, Table 3 should be a figure – see also other comments on additions that have to be made to Table 3 (subfigures, caption etc), to apply these also here.

Answer: We changed tables into figures without table 3 where we put additional porosity measurement results regarding your comment about the lack of that kind of analysis. Also, we put an additional sentence (green-highlighted) where we explained why the porosity level is high in our opinion. The scale indicator has been attached.

We hope our corrections and explanations meets your expectations. We did our best to fit our manuscript for your and other reviewers’ comments. Thank you very much for your valuable and helpful comments.

With kind regards,

Authors

Round 2

Reviewer 1 Report

The paper seems to be corrected properly considering my comments.

Author Response

Dear Reviewer, 

Thank you very much for taking your time and providing valuable comments to improve our paper. 

Yours Sincerely,

Authors

Reviewer 2 Report

The drawing in Figure 2 should have units (mm?) - please add

All other comments have been addressed

Author Response

Dear Reviewer, 

Thank you very much for taking your time and providing valuable comments to improve our paper. Figure 2 has been updated with units.

Yours Sincerely,

Authors